# Injecting structural hints:
# Using language models to study inductive biases in language learning

**Isabel Papadimitriou** and **Dan Jurafsky**
Computer Science Department
Stanford University
{isabelvp,jurafsky}@stanford.edu

## Abstract

Both humans and large language models are able to learn language without explicit structural supervision. What inductive biases make this learning possible? We address this fundamental cognitive question by leveraging transformer language models: we inject inductive bias into language models by pretraining on formally-structured data, and then evaluate the biased learners' ability to learn typologically-diverse natural languages. Our experimental setup creates a testbed for hypotheses about inductive bias in human language learning. We investigate the effect of injecting models with three types of inductive bias: 1) recursive, hierarchical processing, 2) crossing token-token relationships that can't be modeled by context-free grammars, and 3) a Zipfian power-law vocabulary distribution. We show that non-context-free relationships form the best inductive biases. Our study leverages the capabilities of transformer models to run controlled language learning experiments that are not possible to run on humans, and surfaces hypotheses about the structures that facilitate language learning in both humans and machines.

## 1 Introduction

Natural languages are complex and structured systems which humans learn without direct structural supervision. It is a fundamental and open problem in linguistics and cognitive science to understand the inductive biases that make such learning possible: what structural predispositions does a successful language learner need to start with? In this work, we shed light on this cognitive problem using artificial learners. Using our method of structural injection, we causally intervene in transformer language models and manipulate their structural inductive biases, before training on natural language.

We predispose transformers with three structural biases from the cognitive literature: a bias for recursive processing, a bias for keeping track

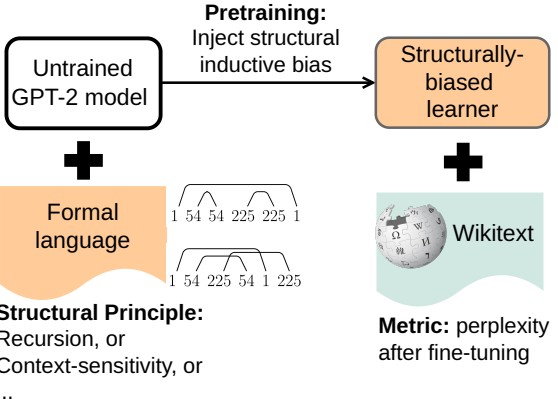

Figure 1: Our method: we take a GPT-2-sized model and pretrain it with a formal language corpus (see Figure 3 for examples). We then take these pretrained models and fine-tune them on Wikipedia data to asses each formal structure as an inductive bias for learning English, Japanese, and Basque.

of context-sensitive dependencies, and a bias for a power-law Zipfian vocabulary distribution. We inject untrained transformer language models with each structural bias by *pretraining on synthetic structured data* and then evaluating language modeling fine-tuning on human languages (English, Japanese, and Basque). Our inquiry is structured around three experimental questions:

- **Experiment 1:** How does an inductive bias for recursion compare to an inductive bias for context-sensitive crossing relationships? (Section 5)

- **Experiment 2:** Is a bias for pure constituent recursion better when mixed with a small amount of tokens that break context-free constituency structure? (Section 6)

- **Experiment 3:** Does a learner biased towards learning a power-law Zipfian vocabulary distribution learn language more effectively? (Section 7)

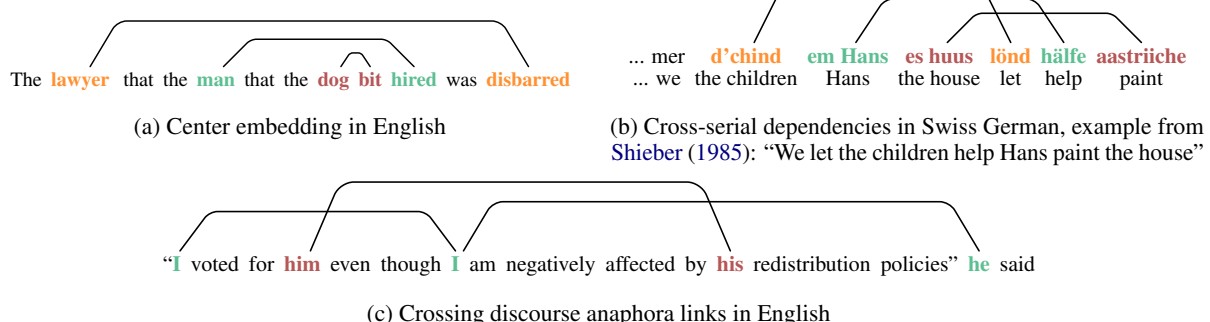

(a) Center embedding in English

(b) Cross-serial dependencies in Swiss German, example from Shieber (1985): "We let the children help Hans paint the house"

(c) Crossing discourse anaphora links in English

Figure 2: Examples of recursive and context-sensitive structures in natural language.

**In Experiment 1, we** *disentangle* **the effects of recursion and context-sensitivity**: both occur in language, but which one is more useful as a sole learning bias? Language is characterized by recursive structures in context-free constituent relationships like those in Figure 2a, and some linguistic theories posit that recursive processing is a crucial (and perhaps the only) inductive bias that makes human language learning possible (Hauser et al., 2002; Chomsky, 1995). However, it is widely hypothesized that human language is mildly context sensitive: while there is recursive structure, there are also non-context-free relationships between tokens, both in syntactic structure (Figure 2b) and in the structure of meaning and discourse relationships (Figure 2c) (Joshi et al., 1990; Stabler, 2010; Shieber, 1985; Steedman, 1990; Frank and Hunter, 2021; Joshi, 1985). We compare the two principles of recursion and non-context-free relationships, and find that an inductive bias for non-context-free crossing dependencies is better for downstream language learning. However, both inductive biases greatly outperform random and regular language controls.

**In Experiment 2, we** *combine* **recursion and context-sensitivity** and ask: is a recursive inductive bias better with slight context-sensitivity? We show that there is a significant improvement in downstream language learning if we add just 1% of a bias for crossing dependencies, breaking the constituent structure of the other 99% recursive bias we give the learner. Even when learners are mostly biased towards recursion, they learn language faster when their bias includes constituent-breaking context-sensitive structures.

**In Experiment 3, we test the effect of inductive biases in vocabulary distribution**: does a bias towards a human-like Zipfian vocabulary distribution (Zipf, 1936) help language learning? Lan-

guage is structured not only in how tokens relate, but also in the structure of the distribution that tokens are drawn from, a cognitive bias that is especially significant for memory-based theories of grammar (Shufaniya and Arnon, 2022; Piantadosi, 2014; Ellis and O'Donnell, 2012). We show that a Zipfian pretraining bias makes models better at downstream language learning even when there is no correspondence between the pretraining and fine-tuning vocabularies.

While our experiments work with computational models, the question that we are examining is about humans: what are the possible inductive biases that make human language learning possible without explicit structural supervision? Using computational models, we can investigate this question through *empirically manipulating the inductive bias of a language learner* — a causal experimental route that is not possible when working with humans. Results from such experiments can act as firstly as proofs of concept, showing how language learning is or isn't possible with different inductive biases. Secondly, such experiments help with hypothesis generation: with structural injection, we can test arbitrary inductive biases in a theory-independent way (see Baroni, 2022; Portelance, 2022; Lappin, 2021; Wilcox et al., 2023; Kauf et al., 2023, for further discussions on the role of neural NLP models in cognitive science). In Section 8 we discuss how our method of leveraging artificial learners to causally investigate inductive biases adds a new direction to the rich prior literature on inductive bias in neural learners.

In summary, our findings indicate that biases for complex token-token interactions, whether these involve recursion or not, form a powerful inductive bias for learning natural language. Crucially, our results are compatible with and can inform a wide

variety of cognitive architectures: models in which structural inductive biases in humans arise from prior statistical learning (Elman, 1996), models in which they arise from other aspects of cognition or communication (Lieven and Tomasello, 2008; Hahn et al., 2020; Gibson et al., 2019), and models in which they are presumed to be innate (Hauser et al., 2002). [1]

# 2 Background: Structural inductive bias in humans

In order to use our experiments as a window into hypotheses about human inductive biases, we look at three families of structural bias from the linguistics and cognitive science literature.

## 2.1 Recursion

One very prominent hypothesis for linguistic inductive bias focuses on recursion: the ability for hierarchically-structured constituents to contain other constituents of the same type, and as such allow for potentially infinitely deep hierarchical structure.

Recursive structures can be described in terms of a context-free language: a grammar with rules of the form $A \rightarrow \beta$, where a non-terminal node $A$ consists of a string $\beta$ that can contain both terminal and non-terminal nodes. Such a phrase-structure grammar is recursive when a non-terminal can contain a string that includes another non-terminal of the same type. For example, one rule describing the language of well-nested parentheses is

$$S \rightarrow ( \ S \ )$$

where any well-formed sentence $S$ can make another well-formed sentence $S$ if it is inserted into a pair of parentheses. Rules of this type allow for infinite nesting.

A canonical linguistic example of recursion is center embedding. In the center embedding example in Figure 2a, a noun phrase ("the man that the dog bit") can be used as a part of another noun-phrase ("the lawyer that [the man that the dog bit] hired"), and this can carry on recursively. Such self-embedding structures, which are attested in human language, are possible with recursive grammars but not in finite-state languages (Chomsky, 1959). The recursion hypothesis for linguistic inductive bias

states that the ability for such constituent recursion is a crucial linguistic inductive bias, constitutes the underlying structural bias for the faculty of language, and that recursion is what distinguishes language from animal communication (Hauser et al., 2002; Chomsky, 1995).

## 2.2 Context-sensitivity

Although context-free grammars allow recursive structure, they are not complex enough to model many attested linguistic effects, which require a non-context-free (i.e., at least context-sensitive) grammar (Joshi et al., 1990; Stabler, 2010; Shieber, 1985; Steedman, 1990; Frank and Hunter, 2021; Joshi, 1985). For example, Shieber (1985) proves that a grammar modeling the unbounded cross-serial dependency structures possible in Swiss German (Fig. 2b) must be non-context-free, and Steedman (1990) shows that gapping effects (sentences like "Harry eats beans, and Fred potatoes") similarly cannot be analyzed by context-free grammars. Looking beyond syntax, reference and discourse structures (Fig. 2c) as well as meaning-based interactions do not abide by context-free restrictions.

While it is broadly hypothesized that both recursion and non-context-freeness exist in natural language, our experiments *empirically disentangle* the two hypotheses. We compare two inductive biases: an inductive bias for recursive processing, and an inductive bias for non-context-free structures that do not have recursion. Non-context-free structures allow for unbounded and crossing interactions between tokens, while recursive grammars only allow tokens to influence each other in specific subtree relationships. To disentangle these biases, we test the effect of an inductive bias where there are very minimal (and non-tree-structured) restrictions on where related tokens can appear (see Section 4.2 and Fig. 3b for details of our formalization).

## 2.3 Vocabulary distribution

We next investigate structural biases in the lexicon: how a cognitive bias towards vocabulary distributions (i.e., which tokens are more likely to appear in a corpus of language) affects learning. We aim to answer: does a bias towards a Zipfian vocabulary distribution (Shufaniya and Arnon, 2022; Piantadosi, 2014) act as a structural bias that aids language learning?

A feature pervasive across human languages is the unbalanced nature of vocabulary distributions. In human languages, some words are very common

---

[1]Code and instructions for running our experiments is at `https://github.com/toizzy/injecting-structural-hints`.

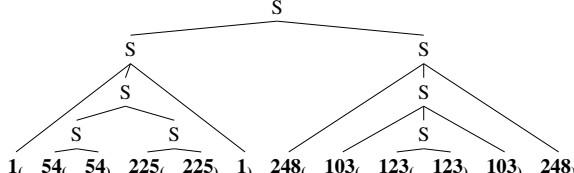

(a) The recursive NEST language. Every S node represents a well-formed NEST substring.

(b) The context-sensitive CROSS language. Edges connect matching parentheses.

**499 472 300 345 272 309 17 15 329 233 9 267 122**

(c) The random language, RAND

**499 472 300 345 272 499 472 300 345 272 309 17 15**

(d) The regular REP language, with repetition length $k = 5$. The repeated block is circled. For our experiments, we use $k = 10$.

Figure 3: The formal languages we use to pretrain our models to give them different structural inductive biases. For simplicity, we represent NEST and CROSS as 250 open tokens with 250 close tokens, while we present RAND and REP with 500 tokens. The vocabulary distributions are the same between all languages if we concatenate the open and close tokens: token $1_)$ in the parentheses languages is equivalent to token 251 in RAND and REP.

(like "the" or "a" in English) and most words are used very rarely, roughly following a power-law distribution. Zipf's law states that the $r$th most common word has frequency proportional to

$$\frac{1}{(r + \beta)^{\alpha}} \quad (1)$$

with $\alpha \approx 1$ and $\beta \approx 2.7$ providing a good empirical estimate for human languages (Zipf, 1936; Mandelbrot, 1953). The literature on explaining and deriving this empirical fact is rich and varied (see Piantadosi, 2014, for a thorough review), while experimental and theoretical works examine the learnability of Zipfian-distributed data (Ellis and O'Donnell, 2012; Lavi-Rotbain and Arnon, 2022; Schuler et al., 2017; Chan et al., 2022). In our work, we examine the effect of a Zipfian inductive on learning language (Experiment 3).

# 3 Method Overview: Pretraining as structural inductive bias

We set up experiments where we control the inductive bias of a language model learner, and examine how inductive biases influence language learning. Our experiments consist of two steps: 1) pretraining a GPT-2-sized model on a formal language to inject a structural bias and 2) fine-tuning on limited natural language data (drawing on our prior methods in Papadimitriou and Jurafsky (2020), see Figure 1 for a depiction). To instill an inductive bias in an artificial language learner, we pretrain untrained transformer language model on synthetic data which exhibits *a single formal structural principle*. After sufficient pretraining on such a corpus, we have a transformer model which has never seen human language, but has successfully learnt to model a type of structure. We can then train this model on natural language data, taking its pretraining to be a structural inductive bias. We fine-tune each model on three different languages (English, Japanese, and Basque). The final performance on each language, measured in perplexity, is an indicator of how learnable the raw language data is when a learner starts with the specific inductive bias.

## 3.1 Implementation Details

For each experiment, we randomly initialize a GPT-2-small model with a max sequence length of 512, and train on one formal language.[2] The specific formal languages we use are described in Section 4. We use a batch size of 512 to train all models, and train for 5,000 steps with 1,000 steps of warmup. Each formal language corpus consists of 1 billion tokens, which comes out to 3,814 batches. Therefore, in training 5,000 steps the model sees each corpus approximately one and a half times.

We fine-tune each pretrained model on three separate languages to measure final test perplexity. We use the the wikitext-103 English dataset (Merity et al., 2017, 103 million tokens), Papadimitriou and Jurafsky (2020)'s Japanese Wikipedia (129 million tokens) and Basque Wikipedia (63 million tokens) corpora. We fine-tune with two epochs on the En-

---

[2]We use GPT-2-small to refer to the model config accessed using Hugging Face `AutoConfig` with key gpt2: 12 transformer layers with 12 attention heads per layer, and a hidden size of 768

glish and Japanese datasets, and 4 epochs on the Basque datasets. Though the three corpora are not controlled for size and train-test difficulty, we do not compare results in order to analyze differences between languages and draw conclusions from this. We instead use the three languages to verify that any claims about language learning and inductive bias that we make are likely cross-linguistic and not English-specific.

Our pretraining languages all have a vocabulary size of 500, which has no correspondences with the 50,257-size vocabulary of GPT-2. To accomodate this new tokenizer, we have to initialize a new word embedding matrix, with 50,257 rows for the model to learn in fine-tuning. We initialize the embedding matrix by randomly sampling with replacement from the rows of the old embedding matrix with 500 rows, following Wu et al. (2023), who show that initializing an embedding matrix for transfer learning by sampling from the old embedding matrix leads to far better transfer learning than random reinitialization even for unrelated vocabularies (see also Hewitt, 2021). To account for the effect of having to relearn the vocabulary, we also add a **control case** where we take the pretrained GPT-2 model and randomly resample the rows of the embedding matrix. To re-learn the embedding matrix, we fine-tune on the natural language corpora. This control appears on the right of our results graphs.

## 4 Pretraining languages

Our experimental method rests on pretraining language models on well-chosen formal languages. Here, we describe the four families of formal languages that we use for pretraining, and present examples of each of the languages in Fig. 3.

### 4.1 Recursive and context free: the nested parentheses language NEST

The first language that we pretrain on is a language of matching nested parentheses, also known as the Dyck language. The vocabulary of this language consists of a set of matched open and closed tokens, and vocabulary size can be set to any even number (for our experiments, we used a total vocabulary size of 500 for all pretraining language). Writing the language from left to right, at each step we randomly pick between two choices: either 1) open a parenthesis ($p = 0.49$) or 2) close the last opened parenthesis ($p = 0.51$). Since we only ever close the most recently-opened parenthesis token, there

will never be any crossing arcs in how we connect parentheses. The nested parenthesis grammar is context-free. An example of this language is shown in Fig. 3a.

### 4.2 Non-context-free: the crossing parentheses language CROSS

The Crossing Parentheses language CROSS is a parentheses language with the same vocabulary of open and close tokens as the NEST language, but where parentheses do not have to be well-nested, but just have to be balanced: opened parentheses must close. As such, pairs of parentheses can interleave and cross. Whereas the NEST language imposes strict limitations to how different tokens can interact (a parenthesis can only close in the space between its parent opening and closing, so there is no ability to see past the top of the stack), in the CROSS a token can have its pair in any location. In order to control the CROSS language to be like the NEST language as much as possible (except for the dependent variable of non-context-free crossing) we add another structural constraint: the *distribution* of distances between open and close tokens is matched to the empirical dependency distance distribution of the NEST language. This way, any difference in the inductive bias is not due to the lengths of dependencies, but due to the structural biases in the pretraining data. This dependency link distribution is what gives the CROSS language structural information for a language model to learn: for every opened parenthesis, there are positions when it is more likely to be closing than others, and modeling this language involves modeling those probabilities.

The crossing parentheses language is not context free. Though every NEST string is a legitimate (but exponentially unlikely) CROSS string, the CROSS language can include other strings, like arbitrarily long cross-serial dependency structures that are known to be non-context-free (Shieber, 1985):

$$(_1 \ (_2 \ (_3 \ \cdots \ (_k \ )_1 \ )_2 \ )_3 \ \cdots \ )_k$$

In the NEST language, arbitrarily deep nesting (and arbitrarily long dependencies) are in the limit possible, though in practice unlikely, and similarly in the CROSS language arbitrarily long dependencies are possible but unlikely. This is roughly analogous to human language, where processes like center embedding and cross-serial dependencies can in theory happen infinitely but in practice are

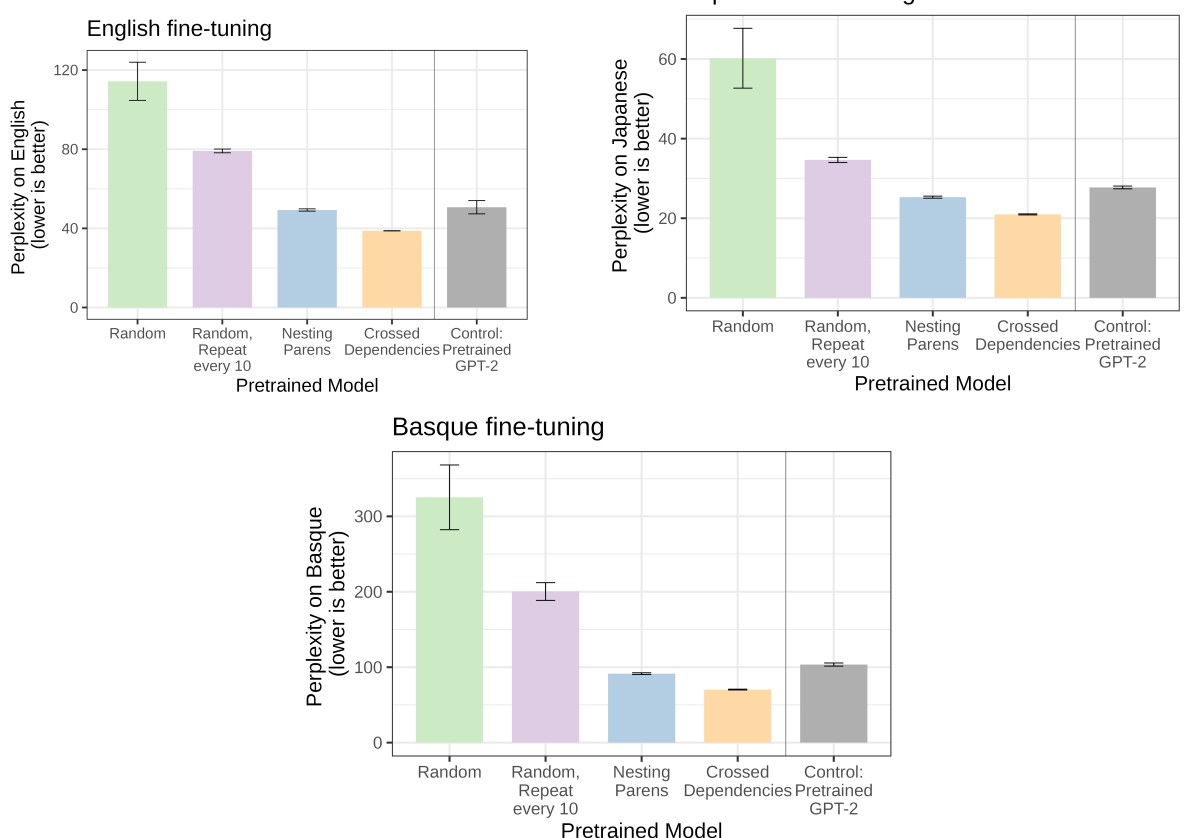

Figure 4: **Results for Experiment 1, CROSS is a better inductive bias than NEST** Each model pretrained with the formal language on the x-axis is evaluated on a wikipedia test set after natural language fine-tuning. Error bars represent 95% confidence intervals over 5 fine-tuning runs with different random seeds. Since we cannot directly compare perplexities between different test sets, we can only compare the ranking of the test conditions. The rank (CROSS is better than NEST is better than REP is better than RAND) is consistent across English, Japanese, and Basque.

limited. This probabilistic weighting doesn't affect our proof of non-context-freeness: the Shieber (1985) proof is independent of the probabilities of these constructions.

### 4.3 Baselines: the Random language and the regular Repetition language

**The Random language RAND** For our RAND baseline, we take the vocabulary of our experimental languages NEST and CROSS, and create a dataset by sampling each token randomly from the vocabulary distribution without any structural limitations.

**The Repetition language REP** We also test a slightly stronger baseline than RAND, the repetition baseline. In the REP language, tokens are placed randomly, like in RAND, but every 10 random tokens are then immediately repeated. Using the repetition language as a baseline lets us control for the effect of there being *any* structure that connects

two tokens. This way, we can better measure the utility of more complex structural inductive biases. Note that REP is not the same as the 'copy language' (the set of arbitrarily long strings followed by their repetition), which is famously context-sensitive. In the case of REP, we're restricting any copied chunk to be exactly $k$ in length, which makes the language regular, indeed subregular.

## 5 Experiment 1: Disentangling recursive and context-sensitive inductive biases

For our first experiment, we compare the fine-tuning perplexity of models pretrained on the NEST language and the CROSS language to each other and to random and regular controls. Our fine-tuning results are shown in Fig. 4. From the fine-tuning perplexity that we get with different structural inductive biases, we can present two findings:

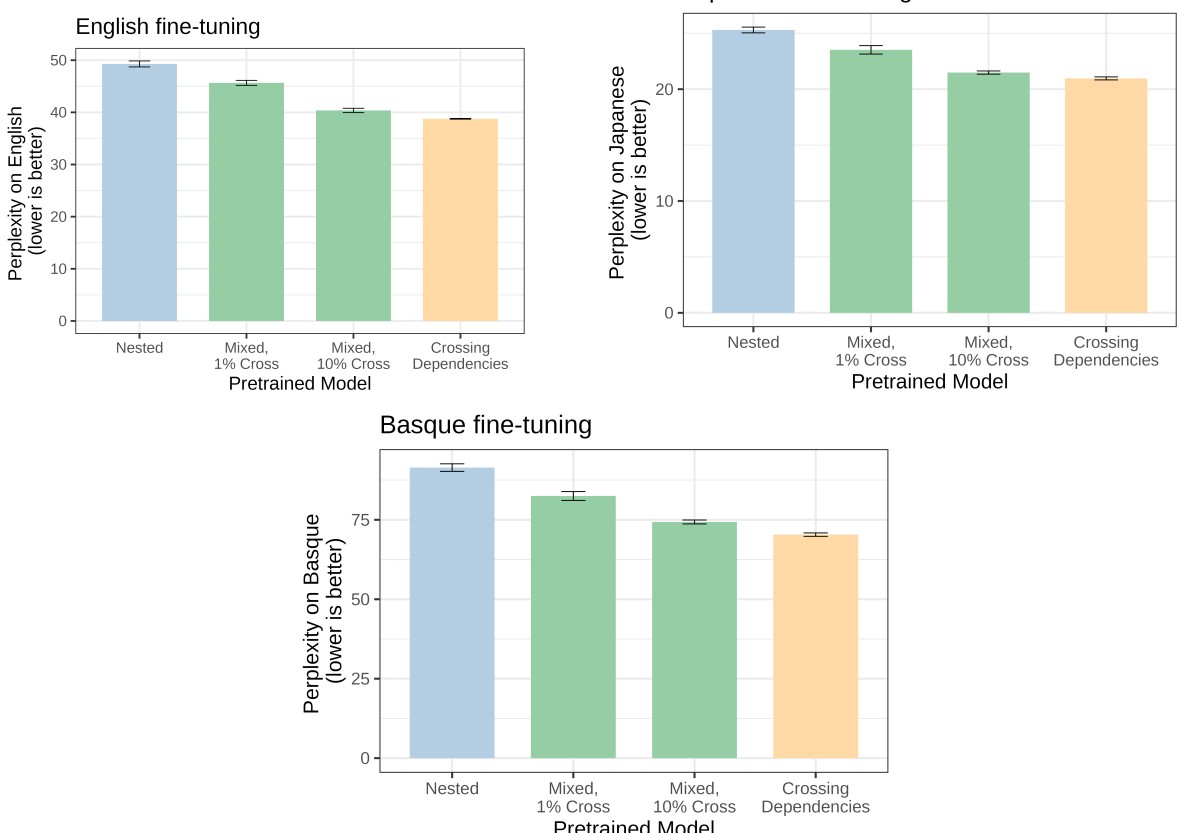

Figure 5: **Results for Experiment 2, even small amounts of non-context-free structures in the inductive bias cause significant improvement in downstream perplexity**. Mixing 1% and 10% of the CROSS language in with the NEST language, which breaks the recursive constituency structure of NEST , causes significant downstream language learning improvements over plain NEST . Error bars represent 95% confidence intervals over 5 fine-tuning runs with different random seeds

***Any*** **structure has a significant effect as an inductive bias**    All of the models with a structural pretraining language fare significantly better than the model with the random, unstructured pretraining language. This especially surprising in the case of the REP language, which has a very simple structure that is similar to RAND. However, our experiments also show that the shallow structure of REP is worse than both of our languages with more complex structural organization. Comparing the performance of the NEST and the CROSS language leads us to our next finding:

**Context sensitivity acts as a better inductive bias than recursion**    Our experiments disentangle the effects of recursive structure and non-context-free linking structure, and show that the non-context-free CROSS language acts as a stronger inductive bias for language learning than the recursive NEST language. Our current methodology does not ascertain which aspects of language learning the

CROSS language is most helpful for, which is an especially fruitful question for future work since context-sensitive linking structures arise in syntax, semantics, and discourse. Understanding the role of different complex structures in influencing language learning in an in-vitro paradigm such as ours shines light on the properties of language as a learnable system under different cognitive structural assumptions.

## 6   Experiment 2: Mixing context-free and non-context-free structures

In Experiment 1, we saw that the non-context-free inductive bias of the CROSS language is more beneficial for downstream language learning than the constituent recursion of the NEST language. However, neither of these extremes encompass the more widely agreed nature of linguistic syntactic structure: likely a recursive grammar with mildly context-sensitive elements.    To examine the

effects of non-context-free elements in otherwise context-free grammars, we create the NEST-MIX-P languages. These languages follow the NEST grammar, except that with probability $p\%$ an 'open' token does not follow the NEST grammar and instead follows the CROSS grammar. This means that we sample a dependency distance and place a 'close' token for it without taking into account the constituent structure of the NEST language. The NEST example from Figure 3a, with one CROSS token added and shown in green, would look like this:

1( 54( 54) 225( 225) 123( 1) 248( 103( 123) 103) 248)

We test two such languages: NEST-MIX-1, with 1% CROSS tokens, and NEST-MIX-10, with 10% CROSS tokens. Our results are shown in Figure 5. Adding 1% CROSS tokens to a NEST language causes it to act as a significantly better inductive bias, with an average improvement of 5 perplexity points, while adding 10% CROSS tokens causes an average downstream improvement of 9 perplexity points. Our results show that even small amounts of context-sensitive inductive bias have a big effect on language learning, aligning with theoretical linguistics results about human language being mildly context sensitive.

## 7 Experiment 3: Zipfian structural bias

How does a bias towards a particular vocabulary distribution affect a language learner? We test the effect of a uniform vocabulary distribution versus a Zipfian vocabulary distribution as a pretrained inductive bias. As shown in Figure 6, a Zipfian pretraining distribution of random tokens predisposes a model with a better language-learning bias.

Crucially, there is no connection or correspondence between the pretraining and fine-tuning vocabularies: the pretraining vocabulary is made up of 500 parentheses tokens (whose frequency ranking is randomly ordered in order to make a Zipfian distribution in the Zipfian case), whereas the GPT-2 tokenizer's fine-tuning vocabulary is over 50K tokens. No token correspondence of specific token information is transferred between pretraining and fine-tuning as the tokens are unrelated and we re-sample the rows of the embedding matrix.

Since there is no correspondence, our vocabulary distribution ablations measure the effect of an abstract bias towards one *type* of vocabulary distribution, rather than the effect of knowing the specific distribution of tokens in a language. Our

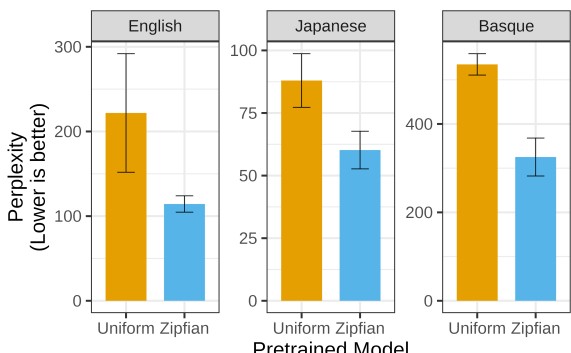

Figure 6: **Results for Experiment 3, a Zipfian inductive bias improves downstream learning**, even when the pretraining and fine-tuning vocabularies are unrelated. We pretrain models with random tokens sampled either from a uniform or a Zipfian distribution.

experiments show that pretraining on a random Zipfian corpus biases a model for better language learning than a uniform corpus, and that a structural bias for vocabulary distribution is encoded beyond the word embedding matrix and more abstractly in network parameters. Due to the lack of any structure beyond vocabulary distribution in the random corpora, neither of the test cases lead to very good downstream language performance. Results around *combining* Zipfian vocabulary with other structural biases like those in Experiment 1 are mixed, and we do not have clear evidence that Zipfian vocabulary has a structural effect that can add on to the effect of structure in data. We discuss these results in more detail in Appendix A.

## 8 Related Work: Inductive bias

Our experiments use transformer models in order to create a controllable testbed for understanding human language learning: by manually varying the inductive bias through pretraining, we can test the effect of different inductive biases in making a learner fit for language learning. Our focus is different, though related, to the focus of much of the work about inductive bias in transformers. While we set the inductive bias of our learners through pretraining, the built-in structural inductive biases of *untrained* deep neural network architectures are unknown and difficult to assess (Baroni, 2022; Warstadt and Bowman, 2022). A rich line of past experiments work towards defining this inductive bias of neural models through empirical methods, looking at how much transformers are biased towards learning hierarchical rules

(Kharitonov and Chaabouni, 2021; Petty and Frank, 2021; Mulligan et al., 2021) or tree-structured processing (Murty et al., 2023), or biases towards different typological linguistic structures (White and Cotterell, 2021; Ravfogel et al., 2019; Ravishankar and Nivre, 2022), and how inductive bias changes with language modeling pretraining (Mueller et al., 2022; Warstadt et al., 2020).

We build on these results by working to influence, rather than measure, the inductive bias of transformers. Our experimental paradigm relies on structural transfer between pretraining and fine-tuning data in unrelated modalities, an effect demonstrated by past research. We showed in previous work how the structures of non-linguistic modalities like music and code can predispose learners for language learning (Papadimitriou and Jurafsky, 2020), work that has been reproduced (Chiang and Lee, 2022; Ri and Tsuruoka, 2022), extended to other modalities like amino acid sequences (Chiang and Lee, 2020), and between language and multiple symbolic tasks (Lu et al., 2022). In a similar line of inquiry, Krishna et al. (2021) show structural transfer from the abstract task structure of summarizing random data to summarizing real natural language passages. The paradigm of structural transfer lets us study the fundamental cognitive issue of inductive bias from an exciting angle: through causal experiments where we *control* the inductive bias of language learners.

## 9 Discussion

Our experiments and methodology provide a new view into the biases that make language learning possible in both human and artificial learners. We use transformer language learners and perform causal interventions that alter their inductive learning biases before training them on natural languages. Our findings show the relative strengths of different structural inductive biases: simple, regular structure is far outperformed by both context-free and non-context-free structural relationships, but a recursive structure is not necessarily the best such complex bias.

We obtain our results from experimenting on artificial language learners, rather than working with humans or analyzing human language. Inductive bias experiments on artificial learners let us easily identify possible hypotheses of structural inductive biases in human cognition. More importantly, since we are working with systems we can

influence (rather than analyzing properties and universals of language) we can test hypotheses that are not dependent on linguistic theory-building. We can therefore explore outside the well-studied hypotheses in the linguistics literature.

Our experiments address questions about the learnability of human languages under different structural inductive biases which have been proposed in the linguistics and cognitive science literature as bases for language. However, experiments such as ours cannot directly prove how the human language system is structurally biased. What such experiments can achieve is to shine light on language as a learnable system as a whole, and produce hypotheses about the relevant structures that could be guiding human language learning. In our case, we find that recursion is indeed a strong inductive bias for language learning. However, our experiments also serve to point out that it's not nesting constituent, structurally-recursive properties that *necessarily* most help language learning. Using the CROSS language, we show that when we take away recursion from the inductive bias, but keep the overall framework of paired tokens that connect in non-trivial ways, we get a better inductive learning bias. Thus, our work serves to showcase possible alternatives to the Hauser et al. (2002) hypothesis that recursion is a necessary bias: other structurally-complex ways of relating tokens may be just as strong. Differences such as those between recursion and non-recursive complex structures are hard to disentangle theoretically. As such, an experimental paradigm such as ours is a helpful step in widely exploring the hypothesis space in questions around human language cognition.

Artificial models of language can never definitively prove facts about human language processing. The models pretrained and fine-tuned in this paper are products of opaque optimization processes, and the results that we get do not necessarily match human learning. Nevertheless, our work serves to identify and examine different hypotheses about human inductive learning biases. We hope that such work can help enrich the hypothesis space over which theoretical argumentation, as well as human subject experiments, are conducted, ultimately leading to a richer understanding of the human language learning process.

## Limitations

As discussed throughout the paper, our method uses language models to better understand language learning and cognitive inductive bias in humans, and this comes with all of the limitations of using computational schematic models of cognition to understand the complex and intractable problem of human cognition. Though strong artificial language learners give us a tool with which to run controllable language learning experiments, experiments on artificial learners do not provide any proof about the actual cognitive processes that happen in humans, and our experiments cannot be taken to provide any such proof. Our results serve to showcase the properties of language as a learnable system under different inductive bias constraints, and can inform hypotheses and theories that are worked out and evaluated on human language learning and human subjects.

A limitation of our experiments is that we only evaluate models on natural language perplexity. Perplexity is a coarse-grained measure, and to expand these epxeriments we plan to further examine *which specific parts* of language production different models are doing better in. Understanding which inductive biases lead to which acquisition patterns, especially with respect to the acquisition of semantic and syntactic patterns, would be a very interesting addition to these findings.

Lastly, our experiments evaluate language learning by fine-tuning on modeling wikitext, which is not similar to the learning environment of human language. Our methodological contribution is independent from the actual data we use, and one way our method could be applied would be in more realistic language acquisition situations, using datasets of child-directed speech for fine-tuning (Warstadt et al., 2023). Understanding the effects of inductive biases in scenarios closer to human language acquisition would be a great next step for this paradigm.

## Ethics Statement

Our experiments address a cognitive question about language learning by running in-vitro experiments on language models. Such methodologies do not provide definitive proof about any human cognitive processes. Instead, such experiments use language models in the way that computational models of cognition are generally used: in order to surface and experiment on interesting cognitive hypotheses. Methodologies such as ours, especially applied to more narrow definitions of learning than language learning, could become harmful if they are taken to provide actionable proof about human learning patterns.

## Acknowledgements

We would like to thank Michael Hahn, Philip Resnik, John Hewitt, the members of the NYU CAP Lab, and the EMNLP anonymous reviewers for enlightening discussions and comments on this paper, as well as all the members of the Jurafsky lab for their consistent and inspiring research support. This research was funded in part by NSF award number IIS-2128145.

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

## A  Combining syntactic structure and vocabulary distribution

In Experiment 3 (Section 7), we showed that a Zipfian vocabulary distribution provides a stronger structural inductive bias than a uniform distribution. Here, we present results in *combining* a Zipfian distribution with the grammatical structural biases in Experiment 1. The results are shown in Figure 7, and do not definitively point one way or the other regarding combining grammatical structure and vocabulary distribution: a Zipfian pretraining distribution creates a stronger bias in some cases and a weaker bias in others. More well-powered and controlled experiments, looking at carefully-chosen grammatical structure to combine with vocabulary distribution, in order to understand the interactions between these two aspects of structural information.

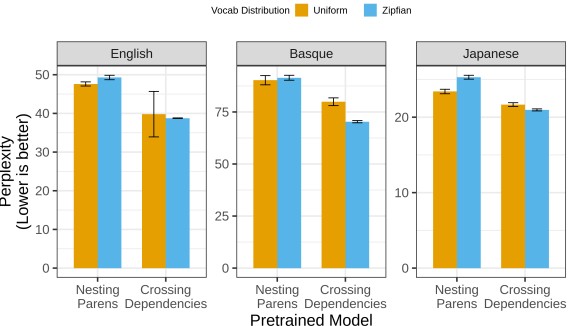

Figure 7: Results for downstream perplexity in all three languages when combining vocabulary distribution biases with ).