# OpenReview forum: "Injecting structural hints: Using language models to study inductive biases in language learning"
_EMNLP/2023/Conference — EMNLP 2023 Findings_

### Official Review · Reviewer_MCzq · 2023-07-26

**Soundness:** 3

**Excitement:**

3: Ambivalent: It has merits (e.g., it reports state-of-the-art results, the idea is nice), but there are key weaknesses (e.g., it describes incremental work), and it can significantly benefit from another round of revision. However, I won't object to accepting it if my co-reviewers champion it.

**Paper Topic And Main Contributions:**

In this work, the authors pretrain transformers on various synthetic languages, and test for transfer to natural language by measuring perplexity. They find that pretraining on formal languages with any amount of structure, including even repetition of sequences of 10 tokens, has a very large on perplexity after fine tuning on the natural corpus, but pretraining on structure that is more linguistically relevant (in particular, cross-serial dependency) is particularly helpful. Likewise, pretraining on a corpus where the vocabulary follows a Zipfian frequency distribution reduces perplexity after fine tuning.

**Questions For The Authors:**

I may have missed something, but the perplexity for the ZIpfian pretraining experiments is much higher than for the earlier, structural experiments - e.g. see the difference between Figure 5 and Figure 6. Why is that?

**Reasons To Accept:**

The methodology of giving models an inductive bias through pretraining on a formal language is simple and convincing. The specific formal languages that the authors use are well-motivated from linguistics, there are good control languages, and the experiments are repeated for three typologically diverse natural languages. The experiments are elegant and clearly presented.

**Reasons To Reject:**

I am not sure I agree with the current framing - e.g. in the first paragraph, the authors suggest that the study can answer the question: "what structural predispositions does a learner need in order for language to be learnable?". In fact, though, what the experiments show is that certain inductive biases help a transformer learn predict the next word better, but it's not clear that those biases are *necessary* to make language learnable, as there's no criterion defined in the paper for when a language has been learned successfully. In fact, we know from other work that LMs trained from scratch on the same amount of data as used in this paper can learn a lot about language (e.g. syntax), so it's clearly not the case that nothing is learnable without the structural inductive biases proposed in this paper.

The argument would be much stronger the evaluation focused on determining if the model acquired specific linguistic phenomena instead of just measuring its perplexity. Indeed, any sort of error analysis or a more qualitative insights into what in particular pretraining on the each formal language helps with would have improved the paper, and perhaps would have helped explain why even the repetition language is so helpful. The repetition language finding suggests that perhaps copying / "induction heads" are playing a central role here. It would be great to see a test of this hypothesis and in general more analysis of the pretrained models, either behavioral (e.g. evaluate them on strings that involve copying) or representationl (e.g. looking at attention heads).

Also because of the coarse grained perplexity-based evaluation it's not so clear what the inductive biases is that the models trained on formal languages end up acquiring (those may not be exactly the same as the inductive biases the authors intend the models to end up with). I find the results in Figure 4, for example, to be quite surprising: how is it that even after training on 100 million tokens of WikiText there is such an enormous effect of the pretraining corpus?

**Reproducibility:**

5: Could easily reproduce the results.

**Reviewer Confidence:**

4: Quite sure. I tried to check the important points carefully. It's unlikely, though conceivable, that I missed something that should affect my ratings.

**Typos Grammar Style And Presentation Improvements:**

The introduction needs to be reorganized in my view - it currently doesn't have a clear structure and goes back and forth between laying out the argument and providing "related work" style lists of references. As I mentioned above I was also missing a crisper motivation for the experiments and what exactly (if anything) they can teach us about how humans learn language.

---

> ### Author Rebuttal · Authors · 2023-08-29
>
> **COMMENT 1: it's clearly not the case that nothing is learnable without the structural inductive biases proposed in this paper.**
>
> This is certainly true, and we did not mean to frame our inductive biases by saying that they give a model a linguistic advantage which the untrained models do not have (we’ll take care to make this clearer in the text). Instead, **we view our inductive bias injection as a way of _controlling_ inductive bias**. Baroni (2021) argues that untrained models, through their architecture and initialization, have inductive biases towards learning some structures over others and are not blank-slate statistical learners (the recent advances in NLP in fact show us that transformers likely have very good linguistic-leaning inductive biases). Our experiments simply put models in a state where we know a great deal about the structures that they’re biased towards learning, rather than knowing nothing, and so we can run controlled experiments like we do. We’ll take care to make this theoretical approach more clear.
>
> **COMMENT 2: The argument would be much stronger the evaluation focused on determining if the model acquired specific linguistic phenomena**
>
> We definitely agree that perplexity is a coarse-grained measure, and for follow-up papers we are working on more fine-grained evaluations. But we believe that our inquiry into inductive bias needs to be done in two steps.  The first step, inthis paper, is to see what kinds of structures aid language learning, using perplexity on naturally occurring text. Our experiments in this step use a small (for language-modeling-scale) fine-tuning dataset to ensure that the learning differences come from the starting inductive bias. Our current paper showcases both expected and unlikely results on the importance of a multifaceted set of grammatical and distributional structures.
>
> The second step, which we are doing in follow-up work, is to **study a broad set of phenomena to understand _where_ the improvements that we see in this first paper come from**. Looking at the multifaceted ways in which language is structured, we want to distinguish the effects that different inductive biases have in downstream linguistic features: syntactic structure, semantic and lexical cohesion, and discourse structure. Previous work in the field thinking about structure in language learning focuses largely on syntactic structure, but structure plays a key role in many linguistic levels (the lexicon, coreference, discourse coherence, etc) so   understanding the linguistic causes of  results requires a broader paradigm which we believe to be a suitable second paper.
>
> In the current paper, we look at perplexity on naturally occurring language, and **we’ve also performed further experiments since submission to look at a simple syntactic phenomenon: agreement**, which we can add to the camera-ready. Performance on simple subject-verb agreement (where we just check if the models have learnt the grammatical system of agreeing verbs to the subject when there aren’t other nouns to confuse them) is very similar to our perplexity results, with CROSS > NEST > REP > RAND.
>
>
> | Model | Accuracy |
> |-----------|------------|
> | Random | 74.8 |
> | Regular | 75.5 |
> | Nest | 86.9 |
> | Cross | 90.0 |
> | Control: GPT resample embs | 79.3 |
> | Control: off-the-shelf GPT-2 | 95.8 |
>
> Follow-up work looking at complex linguistic abilities would require a slightly different paradigm:  we would need a much larger fine-tuning dataset in order to induce consistent performance on the finer points of grammaticality. With our current fine-tuning size, we ran some experiments testing more complex syntactic phenomena from SyntaxGym, and all models (including the off-the-shelf GPT-2 control with shuffled embeddings) perform close to chance, and so we cannot draw meaningful conclusions. The failure of our control case suggests that the fine-tuning dataset size is insufficient to test such subtleties, as the lexicon is not developed enough after wikitext-103 training to be able to showcase underlying structural subtleties in difficult natural environments. A follow-up paper with a much larger fine-tuning set will include experiments on more sophisticated grammatical, semantic, and discourse phenomena, and would also need ways to differentiate the structure inserted in pretraining from structure learned during fine-tuning,  and we believe this large set of experiments to be out of scope for this first paper.
>
> (we also wrote this last section to reviewer Wikh, who had a similar question)
>
>
> **COMMENT 3 Why is the Zipfian perplexity higher than the other experiments?**
>
> The Zipfian results that we report in the main paper are with RAND languages, where the only information comes from the distribution that the tokens are being randomly sampled from. So they should be compared to the RAND language perplexity in Experiment 1. We also test the effect of combining distribution and structure and report these results in the Appendix, Figure 9. We can promote this and some further discussion to the main text in the camera-ready
>
> Thanks for all of the smaller comments as well, it’s very valuable to see your view on the presentation of the work and we will work to adopt a lot of this in rewriting!

---

### Official Review · Reviewer_A6Y3 · 2023-08-04

**Soundness:** 4

**Excitement:**

3: Ambivalent: It has merits (e.g., it reports state-of-the-art results, the idea is nice), but there are key weaknesses (e.g., it describes incremental work), and it can significantly benefit from another round of revision. However, I won't object to accepting it if my co-reviewers champion it.

**Paper Topic And Main Contributions:**

This paper experiments with pretraining a transformer language model on a formal language, then fine-tuning on a natural language. The formal pretraining languages are NEST (= the Dyck language), and CROSS, the language of parentheses that are balanced but not nested. The natural languages are English, Japanese, and Basque. The experiments show that NEST is better than random training data, and CROSS is better than NEST. Furthermore, pretraining with a Zipfian distribution also helps.

**Questions For The Authors:**

A. Is it possible to say more precisely what you mean by "structure" (line 463)? It's especially jarring to say that RAND has no structure (466) but REP has a structure similar to RAND (469).

B. In Experiments 1 and 2, how many pairs of parentheses do NEST and CROSS use?

C. In Experiment 3, are the parentheses nested and/or balanced? I would think neither, but if so, I can't think of any reason why the alphabet consists of parentheses.

**Reasons To Accept:**

The findings that NEST is better than RAND and that the Zipfian distribution helps are interesting and expected; the finding that CROSS is better than NEST is interesting and unexpected.



**Reasons To Reject:**

Although the findings are interesting, the paper does not make much effort to investigate why the effects occur, or even offer any guesses about why CROSS is better than NEST.

Not reasons to reject, but things that can be improved:

line 369: Please clarify whether the alphabet can contain multiple pairs of parentheses (sounds like yes).

line 416: The fact that this language contains cross-serial dependencies isn't enough to prove that it is not context-free. ($\Sigma^*$ contains everything, but is context-free.) You have to intersect the language with the regular language ${(_1}^* {(_2}^* {)_1}^* {)_2}^*$ to obtain $\{ {(_1}^m {(_2}^n {)_1}^m {)_2}^n \mid m,n\ge 0 \}$, which is not context-free.




**Reproducibility:**

4: Could mostly reproduce the results, but there may be some variation because of sample variance or minor variations in their interpretation of the protocol or method.

**Reviewer Confidence:**

4: Quite sure. I tried to check the important points carefully. It's unlikely, though conceivable, that I missed something that should affect my ratings.

---

> ### Author Rebuttal · Authors · 2023-08-29
>
> Thanks for the encouraging review and the interesting points you bring up! It’s been a very helpful read for framing our paper.
>
> **COMMENT 1  The paper does not make much effort to investigate why the effects occur, or even offer any guesses**
>
> Thanks for your comment, this is a question that we’re very interested in! With Experiment 2, we run a first  investigation to understand why CROSS may aid language learning. Our hypothesis that we want to test is that the CROSS performance is caused by the fact that the language is context-sensitive, rather than other artifacts that arise because of the specific structure of the CROSS language. The NEST-mix-p languages test this hypothesis by adding occasional non-context-free parenthesis pairs to the NEST language, creating a context-sensitive language that has different commonly-seen patterns than the CROSS language. Our results show that the addition of such pairs causes a significant increase in performance, showing that the context-sensitivity of CROSS can explain a large part of the increase in performance. We’ll work to make the purpose and hypothesis testing of Experiment 2 more clear, as we realize that it’s not very well signaled in the text currently.
>
> In terms of more broad reasons or guesses: we believe that context-sensitive structures are abundant in language, though many may not exactly parallel the specific structure of the CROSS language. For example, a word with certain semantics appearing somewhere changes the probability distribution of what words will appear around it, the same discourse referents keep being mentioned by pronoun and name, a conjunction signals a direct correspondence between phrases on both sides, and so many more. The CROSS language equips a model to learn these language structures at many levels, and that may be more useful than the NEST model that more reflects syntax. In follow-up work, we plan to set up a paradigm to distinguish between the different pre-training languages and the kinds of linguistic processes they help with.
>
> **COMMENT 2 Why would we say that the REP language has structure similar to RAND?**
>
> This was a slip-up in phrasing, thanks for catching this! By saying this, we meant that the REP language had very minimal structure, and so was most similar to the RAND language which had none. We would broadly define structure as the complex and simple ways in which tokens in a language are not independent, both in their position and in their identity. For example, a Dyck language like NEST has a complex structure where one pair of parentheses can dictate where many other ones can close, because the language’s structure prevents intersection.
>
> **COMMENT 3 In Experiments 1 and 2, how many pairs of parentheses do NEST and CROSS use?**
>
> The vocabulary consists of 250 different types of parentheses. We mention this in a caption but not in the main text, which is an omission of ours – we’ll be sure to fix this
>
> **COMMENT 4  In Experiment 3, are the parentheses nested and/or balanced?**
>
> The results in Figure 6 are comparing Zipfian and Uniform versions of the RAND language. Further results combining the structural languages of Experiment 1 with Zipfian distribution effects are in Figure 9 in the appendix. We’ll make this more clear in the prose, because now you’re right that it’s not well-signaled. We use the same vocabulary for all experiments, so the same tokens can act like matching pairs in the languages that have matching, and just like tokens in the languages that don’t require it.
>
> Thanks a lot for this review, and for helping us clarify our points, we’ll take care to take your comments into account!

---

### Official Review · Reviewer_wiKh · 2023-08-05

**Typos Grammar Style And Presentation Improvements:** 1. In Section 6, I was a bit confused…
**Soundness:** 3

**Excitement:**

3: Ambivalent: It has merits (e.g., it reports state-of-the-art results, the idea is nice), but there are key weaknesses (e.g., it describes incremental work), and it can significantly benefit from another round of revision. However, I won't object to accepting it if my co-reviewers champion it.

**Paper Topic And Main Contributions:**

This article examines three types of inductive biases that potentially aid language learning in humans, by exposing language models to data capturing those biases, and measuring whether that, in fact, has a positive contribution to language learning. The three biases examined are recursion, context-sensitive language and a Zipfian distribution over the vocabulary.

After an extensive background section (2) discusses the inductive biases, section 3 goes into the experimental setup: a GPT-2 sized model is pretrained on a formal language meant to inject the inductive bias (focusing on only one principle at a time), and afterwards the model is briefly finetuned on natural language data in 1 of 3 languages. Special care is taken to ensure that the embeddings of the natural language are of adequate quality: they are initialised using randomly sampled pretraining tokens.

Section 4 details 4 formal languages used in pretraining:
1. NEST: a recursive and context-free Dyck language
2. CROSS: using brackets similar to 1, but with crossing dependencies. All opening brackets require a corresponding closing bracket.
3. RAND: randomly sampled brackets without any limitations.
4. REP: randomly sampled strings of 10 tokens where after each 10 tokens, they are repeated immediately

Then, 3 experiments are conducted:
1. Experiment 1 compares the perplexity on English/Japanse/Basque of models pretrained on those 4 languages, observing that CROSS < NEST < REP < RAND. Hence, context sensitivity acts as a better inductive bias than recursion.
2. Experiment 2 introduces a third formal language, NEST-MIX-p that mixes NEST with some crossed dependencies, leading to improved performance compared to NEST (but not CROSS). The authors conclude that this aligns with natural language being mildly context-sensitive.
3. Experiment 3 compares a uniform token distribution to a Zipfian token distribution, suggesting that the latter leads to strong decreases in perplexity for English, Japanese and Basque.

In particular, CROSS being the most advantageous language is an interesting finding, because it suggests recursion is not necessary for a strong inductive bias.

**Questions For The Authors:**

1. Could you please re-explain the control case from line 353 to me? Is the pretrained GPT-2 model the one pretrained by you (in which case… isn’t the control case the same as the actual experiments you are running?) or the “real” GPT-2? Or are the embeddings in the control case randomly sampled from e.g. a  normal distribution? I found this explanation quite confusing. (based on the graphs I think it’s the original GPT-2, but please explain nonetheless)
2. The Wikitext test set is an iid test set, I am assuming. Recursion has been widely discussed as potentially allowing generalisation to more complex structures in compositional generalisation or simply generalisation to longer lengths. Have you performed any experiments using non-iid test sets like that? I’d be very curious about the effect of CROSS and NEST in a setup like that. After all, humans don’t learn language in an iid train-test setup.
3. Could you elaborate on why you think CROSS outperforms the remaining languages?

**Reasons To Accept:**

- The paper presents a method to test inductive biases for language learning that uses formal languages with desirable properties (being context-free & having recursion vs. being context sensitive), and exposes a language model to those formal languages. If pretraining on the formal language improves performance on natural language afterwards, that suggests that these are inductive biases for language learning (and potentially even language learning in humans). The paper successfully confirms that (given their experimental setup) introducing recursion and context-sensitivity improves language learning, which is an interesting find to the subcommunity of *ACL conferences interested in (psycho)linguistics.
- The paper also confirms that biasing models towards Zipfian token distributions improves perplexity on natural language, which is similarly an interesting find.
- Apart from the interesting findings, the paper's formal languages and data could be a useful resource for future work.

**Reasons To Reject:**

- [Major] Even though the introduction of the languages and the pretraining procedure are appealing, the experimentation with the resulting models is quite minimal. Language learning is equated to perplexity in three languages. No further downstream tasks are considered, and no challenging train-test deviations are discussed. Particularly given the prominent role recursion is supposed to play in humans' capability to treat language in a systematic and productive way (which is not very well evaluated using just the perplexity on the WikiText test set), this limits the conclusions that can be drawn about the atypical finding that CROSS outperforms NEST(-MIX-p). Additionally, the results are just based on one model architecture, so the reader is left wondering how robust these results are.
- [Minor] The 3rd experiment, testing the Zipfian distribution, seems a little out of place, considering that it evaluates structure over a vocabulary instead of grammatical structure. The paper could present a stronger argument by solely focusing on exp 1 & 2 but extending the experimentation or the discussion of the results.
- [Minor] The article is a bit out-of-balance in general, as it discusses the background almost too elaborately. What recursion is, what context-free / context-sensitive languages are, and what a Zipfian distribution is, is quite well-known in the *ACL community and that text could be shortened significantly. As is, the article might be better suited to a different venue, such as a workshop.

**Reproducibility:**

4: Could mostly reproduce the results, but there may be some variation because of sample variance or minor variations in their interpretation of the protocol or method.

**Reviewer Confidence:**

3: Pretty sure, but there's a chance I missed something. Although I have a good feel for this area in general, I did not carefully check the paper's details, e.g., the math, experimental design, or novelty.

---

> ### Author Rebuttal · Authors · 2023-08-29
>
> Thanks for this review! It’s really thorough and your comments are very useful to think through, and encouraging about the direction of this work. Below we discuss some of the main points brought up:
>
> **COMMENT 1 No further downstream tasks are considered, have you performed any experiments using non-iid test sets?**
>
> We definitely agree that perplexity is a coarse-grained measure, and for follow-up papers we are working on more fine-grained evaluations. But we believe that our inquiry into inductive bias needs to be done in two steps.  The first step, inthis paper, is to see what kinds of structures aid language learning, using perplexity on naturally occurring text. Our experiments in this step use a small (for language-modeling-scale) fine-tuning dataset to ensure that the learning differences come from the starting inductive bias. Our current paper showcases both expected and unlikely results on the importance of a multifaceted set of grammatical and distributional structures.
>
> The second step, which we are doing in follow-up work, is to **study a broad set of phenomena to understand _where_ the improvements that we see in this first paper come from**. Looking at the multifaceted ways in which language is structured, we want to distinguish the effects that different inductive biases have in downstream linguistic features: syntactic structure, semantic and lexical cohesion, and discourse structure. Previous work in the field thinking about structure in language learning focuses largely on syntactic structure, but structure plays a key role in many linguistic levels (the lexicon, coreference, discourse coherence, etc) so   understanding the linguistic causes of  results requires a broader paradigm which we believe to be a suitable second paper.
>
> In the current paper, we look at perplexity on naturally occurring language, and **we’ve also performed further experiments since submission to look at a simple syntactic phenomenon: agreement**, which we can add to the camera-ready. Performance on simple subject-verb agreement (where we just check if the models have learnt the grammatical system of agreeing verbs to the subject when there aren’t other nouns to confuse them) is very similar to our perplexity results, with CROSS > NEST > REP > RAND.
>
>
> | Model | Accuracy |
> |-----------|------------|
> | Random | 74.8 |
> | Regular | 75.5 |
> | Nest | 86.9 |
> | Cross | 90.0 |
> | Control: GPT resample embs | 79.3 |
> | Control: off-the-shelf GPT-2 | 95.8 |
>
> Follow-up work looking at complex linguistic abilities would require a slightly different paradigm:  we would need a much larger fine-tuning dataset in order to induce consistent performance on the finer points of grammaticality. With our current fine-tuning size, we ran some experiments testing more complex syntactic phenomena from SyntaxGym, and all models (including the off-the-shelf GPT-2 control with shuffled embeddings) perform close to chance, and so we cannot draw meaningful conclusions. The failure of our control case suggests that the fine-tuning dataset size is insufficient to test such subtleties, as the lexicon is not developed enough after wikitext-103 training to be able to showcase underlying structural subtleties in difficult natural environments. A follow-up paper with a much larger fine-tuning set will include experiments on more sophisticated grammatical, semantic, and discourse phenomena, and would also need ways to differentiate the structure inserted in pretraining from structure learned during fine-tuning,  and we believe this large set of experiments to be out of scope for this first paper.
>
> (we also wrote this to reviewer MCqz, who had a similar question)
>
> **COMMENT 2 testing the Zipfian distribution seems a little out of place**
>
> Thanks for this comment, we will try to change the framing a bit to make the change seem less jarring.  Grammatical structure and vocabulary distribution are the two most widely-studied linguistic structural biases in the literature. We include the Zipf section and experiments in order to make our results more complete, and to deal with cognitive linguistic inductive biases broadly and at many levels. We believe that it’s important to investigate both of these levels of structural biases in order to truly investigate the possibility of what biases aid language learning, and doing so doesn’t take away from the more traditional grammatical structure experiments.
>
>
> **COMMENT 3 Could you please re-explain the control case from line 353?**
>
> It’s as you suggest, our baseline is taking the real GPT-2 from the huggingface hub, but re-sampling the embedding mataric. We use this model as a baseline because we can assume this model has close to the best possible inductive bias for learning English – it already knows English. We shuffle the rows of the embedding matrix of this model in order to get a baseline for how well we can do with a very good inductive bias but no lexical information, which gives us an upper bound for how much a synthetic bias (with no lexical relation to language) can help. This baseline showcases the importance of vocabulary – there is only so much that training on a small dataset can give a model since the vocabulary is sparsely updated.
> You’re right, reading over our text it’s clear that this should have been phrased more clearly and we’ll take care to amend it.
>
> **COMMENT 4 Could you elaborate on why you think CROSS outperforms the remaining languages?**
>
> NEST-style recursion is a well-studied aspect of syntax, underlying many grammatical structures. We hypothesize that the success of CROSS-style intersecting dependencies lie in that they are relevant in many aspects of language, both in syntax but also in semantic and discourse coherence: a word with certain semantics appearing somewhere changes the probability distribution of what words will appear around it, the same discourse referents keep being mentioned by pronouns and name, a conjunction signals a direct correspondence between phrases on both sides, and so many more. Our experiments show the possible importance in having an inductive bias that serves language learning across many levels, beyond syntax. Follow-up work will focus on disentangling the effect of different pretraining structures on different levels of language, from syntactic structure to semantic cohesion and discourse and reference.
> Thanks a lot for your smaller comments as well, they are very useful and we’ll take care to fix up the paper in those directions. And thank you very much for the thoughtfulness and thoroughness of your review.

---

### Meta-Review · Area_Chair_ErzF · 2023-09-19

**Recommendation:** 4

**Metareview:**

The paper is sound and addresses problems at the core of this track.

---

### Decision · Program_Chairs · 2023-10-07

**Decision:**

Accept-Findings

**Comment:**

The paper is sound and addresses problems at the core of this track.